# Transcriptional Changes in Radiation-Induced Lung Injury: A Comparative Analysis of Two Radiation Doses for Preclinical Research

**DOI:** 10.3390/ijms25073766

**Published:** 2024-03-28

**Authors:** Mohamed El-Agamy Farh, Hyun-Jin Kim, Sang-Yeon Kim, Jae-Hee Lee, Hajeong Lee, Ronglan Cui, Soorim Han, Dong Wook Kim, Sunjoo Park, Yoon-Jin Lee, Yun-Sil Lee, Insuk Sohn, Jaeho Cho

**Affiliations:** 1Department of Radiation Oncology, Yonsei University College of Medicine, Seoul 03722, Republic of Korea; mafarh@arontier.co (M.E.-A.F.); hope0211@hotmail.com (H.-J.K.); sysykim@yuhs.ac (S.-Y.K.); blue007520@yuhs.ac (J.-H.L.); hl688@yuhs.ac (H.L.); choiyr322@yuhs.ac (R.C.); soorim77@yuhs.ac (S.H.); kdw1026@yuhs.ac (D.W.K.); sunpark@yuhs.ac (S.P.); 2Drug Development Team, ARONTIER, Co., Ltd., Seoul 06735, Republic of Korea; issohn@arontier.co; 3Korea Institute of Radiological and Medical Science, Seoul 01812, Republic of Korea; yjlee8@kirams.re.kr; 4Graduate School of Pharmaceutical Science, Ewha Womans University, Seoul 03760, Republic of Korea; yslee0425@ewha.ac.kr

**Keywords:** radiation-induced lung injury, stereotactic body radiation therapy, 65 Gy and 75 Gy, inflammation and fibrosis stages, micro-computed tomography, histopathological analysis, RNA sequencing

## Abstract

In a recent stereotactic body radiation therapy animal model, radiation pneumonitis and radiation pulmonary fibrosis were observed at around 2 and 6 weeks, respectively. However, the molecular signature of this model remains unclear. This study aimed to examine the molecular characteristics at these two stages using RNA-seq analysis. Transcriptomic profiling revealed distinct transcriptional patterns for each stage. Inflammatory response and immune cell activation were involved in both stages. Cell cycle processes and response to type II interferons were observed during the inflammation stage. Extracellular matrix organization and immunoglobulin production were noted during the fibrosis stage. To investigate the impact of a 10 Gy difference on fibrosis progression, doses of 45, 55, and 65 Gy were tested. A dose of 65 Gy was selected and compared with 75 Gy. The 65 Gy dose induced inflammation and fibrosis as well as the 75 Gy dose, but with reduced lung damage, fewer inflammatory cells, and decreased collagen deposition, particularly during the inflammation stage. Transcriptomic analysis revealed significant overlap, but differences were observed and clarified in Gene Ontology and KEGG pathway analysis, potentially influenced by changes in interferon-gamma-mediated lipid metabolism. This suggests the suitability of 65 Gy for future preclinical basic and pharmaceutical research connected with radiation-induced lung injury.

## 1. Introduction

Lung cancer radiotherapy can cause damage to the surrounding normal tissues, known as radiation-induced lung injury (RILI). This injury can manifest as acute and reversible pneumonitis or, in severe cases, as chronic irreversible lung fibrosis [1]. While RILI is seldom fatal, the long-term fibrotic damage resulting from ionizing radiation (IR) exposure can lead to reduced pulmonary function, including hollow organ stenosis, compromised gaseous diffusion, and loss of tissue compliance, ultimately impacting the patient’s quality of life [2,3].

The importance of establishing a suitable small-animal model for studying RILI to accurately replicate the pathological manifestations of the disease and facilitate mechanistic investigations and therapeutic interventions has been underscored. However, a significant challenge in this endeavor lies in devising a system capable of precisely delivering IR doses to the lungs of small animals in a timely manner. Conventional models, which involve whole-lung irradiation at low doses (10–20 Gy), have been acknowledged for an extended period but suffer from time-intensive protocols. The induction of inflammation typically requires approximately four weeks, while the development of fibrosis necessitates 4 to 6 months [4,5]. To overcome these limitations, the adaptation of modern radiotherapy techniques, such as stereotactic radiation therapy (SBRT), to the scale of small animals is imperative. Remarkable progress has been made in small-animal modeling in radiation biology by a group of scientists who have successfully implemented high-dose radiation targeting specific lung regions, consistently inducing advanced fibrotic changes within a couple of months [6]. This breakthrough has opened up novel avenues for comprehensive research to elucidate the underlying mechanisms, pathogenesis, and potential therapeutic strategies for RILI.

In previous studies, 75 to 90 Gy was selected as the main experimental radiation dose, through which lung fibrosis developed within 2 months after IR exposure. Following the selection of 75 Gy from a series of publications, the pathogenesis of RILI was investigated at different levels (tissue, protein, and molecular levels), followed by an investigation into specific mechanisms, including epithelial–mesenchymal transition, endothelial–mesenchymal transition, and fibrosis-specific mechanisms [7,8]. However, as our research progressed to the drug development stage, despite our reliance on the 75 Gy IR dose, we became concerned that excessive radiation damage could carry a risk of underestimating the effectiveness of the candidate drug, especially at the early stage of disease development. Therefore, it was necessary to examine small volume-beamed lower doses in order to find a more appropriate dose for the RILI model.

Recent advancements in genome-wide transcriptome analyses such as microarrays and RNA sequencing (RNA-seq) have revolutionized our ability to explore the comprehensive transcriptomic profile of irradiated and non-irradiated lungs. These techniques have facilitated the identification of radiation-specific genes and pathways that hold potential as radiation-specific biomarkers or therapeutic targets for the development of RILI treatments [5,9,10,11]. Additionally, transcriptomic analysis has emerged as a crucial component in diagnostic research for RILI [6]. For instance, in the early stages of developing the modern RILI model, a comparative microarray study was conducted to examine the transcriptional differences between an innovative radiation methodology utilizing a high dose of 90 Gy and the conventional approach involving whole thoracic-exposed IR. The study revealed distinct molecular signatures for each model, despite substantial convergence at the morphological and cellular levels [6]. These findings highlight the importance of transcriptomic analysis in elucidating the unique molecular characteristics associated with different radiation exposure techniques and doses and their implications in RILI research.

In this study, we aimed to explore the molecular characteristics of 75 Gy-irradiated lungs of mice. Furthermore, for the purpose of animal model optimization, irradiation was performed at doses (65, 55, and 45 Gy) reduced from 75 Gy by 10 Gy, and it was confirmed that the lowest dose that consistently caused fibrosis within 2 months was 65 Gy. We investigated whether this 10 Gy dose reduction would make any difference in imaging, histopathology, and molecular signatures based on transcriptomic analysis in the RILI model and assessed its effects on RILI model suitability. This study highlights—for the first time—the dynamical changes in animal transcriptomes upon exposure to different focally applied high radiation doses. In addition, it serves as the stepstone towards appropriate preclinical studies connected with RILI by providing a more optimized animal model.

## 2. Results

### 2.1. Transcriptomic Analysis of 75 Gy-Based Small-Animal Model

In a series of previous studies, we found that 75 Gy was capable of inducing the two etiological scenarios associated with RILI pathogenesis, including the early pathological stage, pneumonitis, and the late stage, fibrosis. This was evidenced by clinical and histological examinations; however, unlike the previously applied higher dose (90 Gy), the molecular alterations that result from exposure to 75 Gy are unclear. Following dose application, we isolated total RNA after 2 and 6 weeks for RNA sequencing and analysis. RNA-seq and alignment data for the 75 Gy applied dose are summarized in Table 1. To understand the similarity and variation between the RNA profiles of control (non-irradiated) and 75 Gy-irradiated samples, as well as between the 2- and 6-week samples, we normalized raw reads and interpreted them using multidimensional scaling (MDS) and heatmap correlation plots. In the MDS analysis, samples were aggregated into principal components (PCs) based on the time and condition. The RNA of non-irradiated samples clustered away from that of the irradiated samples. Similarly, the RNA from 2-week samples was clustered separately from that of 6-week samples (Figure 1a). Pearson correlation revealed two major clusters between non-irradiated and irradiated groups, which were then sub-clustered to segregated 2- and 6-week samples (Figure 1b). In general, these data indicated the molecular distinction of 75 Gy-irradiated tissues and further highlighted molecular differences between the inflammation and fibrosis stages induced by exposure to 75 Gy.

Next, we identified the differentially expressed genes (DEGs) induced by 75 Gy at the inflammation and fibrosis stages using edgeR. As shown in Figure 1c, the total number of DEGs, including up- and downregulated genes, was higher in the 6-week samples than in the 2-week samples. The total numbers of DEGs in 2-week and 6-week samples were 1023 (742 up- and 281 downregulated) and 1305 (992 up- and 313 downregulated), respectively. Using a Venn diagram, we were able to obtain more resolution on overlapping and distinct DEGs of the two groups. We observed that 850 DEGs, accounting for nearly 31% of the total DEGs, were identified as overlapping DEGs. We named this group of DEGs IRiDEGs. For stage-specific DEGs, the inflammation-specific group (termed IsDEGs) had 573 DEGs, accounting for 21% of the total, whereas 1290 fibrosis-specific DEGs (termed FsDEGs) were identified, accounting for approximately 47% of the total DEGs. These results indicated that the number of FsDEGs was higher than that of the IsDEGs (Figure 1d).

To determine the primary functional process driven by each of the common and stage-specific DEGs, we performed Gene Ontology (GO) enrichment and pathway analysis using the webpage gene annotation and analysis platform Metascape, in which analyses were carried out using different ontology resources including GO biological processes, Kyoto Encyclopedia of Genes and Genomes (KEGG) pathways, reactome gene sets, CORUM, TRRUST, PaGenBase, WikiPathways, and PANTHER pathways. The results suggested that inflammatory response, leukocyte chemotaxis, positive regulation of response to external stimulus, regulation of immune system process, and neutrophil degranulation were the most significant ontologies and pathways in IRiDEGs (Figure 1e). Mitotic cell cycle, innate immune response, response to type II interferons, and positive regulation of cytokine production were the most significantly enriched ontologies in IsDEGs (Figure 1f), and immunoglobin production, circulatory system process, regulation of epithelial cell proliferation, calcium signaling pathway, regulation of cell activation, and cell–cell adhesion were the most significantly enriched ontologies and pathways in FsDEGs (Figure 1g).

### 2.2. Lowest Single Irradiation Dose to Induce RILF within 2 Months

We lowered the radiation dose from 75 Gy at 10 Gy intervals to 65 Gy, 55 Gy, and 45 Gy to find the lowest dose at which histologically distinct fibrosis was reproducibly induced 6 weeks after irradiation. The characteristic ring-like boundary, accompanied by a white-colored adjacent area typically observed at the beamed location following exposure to 75 Gy, consistently developed after exposure to 65 Gy. However, this phenotype was not consistently observed in animals exposed to IR at 55 Gy and 45 Gy. Histopathological analysis corroborated these observations, as the damaged area surrounding the beamed location was most evident at the lowest microscopic magnification in lung tissue slides from animals exposed to 65 Gy. In contrast, minimal damage was observed in lung tissue slides from animals exposed to 55 Gy and 45 Gy (Figure 2a). Moreover, the evaluation of collagen deposition through image analysis of slides stained with Masson’s trichrome revealed the highest accumulation in lung tissue slides from animals exposed to 65 Gy, while moderate and low accumulation were observed in lung tissue slides from animals exposed to 55 Gy and 45 Gy, respectively (Figure 2b). Based on these findings, we concluded that a radiation dose of 65 Gy was suitable for inducing a 2-month-based fibrotic reaction and selected this dose for subsequent comparative analysis with 75 Gy.

### 2.3. Comparative Imaging and Histopathological Analysis of 65 Gy and 75 Gy Models

We proceeded to validate the inflammatory and fibrotic effects of the 65 Gy radiation dose by comparing it closely with the previously established dose of 75 Gy. Initially, we observed minimal differences between the 65 Gy and 75 Gy doses in terms of gross observation. Both doses resulted in the formation of a ring-like boundary at the beamed location, accompanied by a white-colored adjacent area after 2 weeks. A more pronounced phenotype at the beamed location was observed in the lungs of irradiated mice after 6 weeks in both dose groups.

To further evaluate the irradiated lungs, we performed 2D and 3D image-based analysis using micro-computed tomography (micro-CT). This analysis indicated a clear manifestation of RILI in both the 2-week and 6-week irradiated mice compared to that in the control group. In both the 75 Gy- and 65 Gy-irradiated lungs, pulmonary consolidation was more commonly observed on the lesion side of the left lung (grey area) than in the control group, particularly in the 6-week group. However, the extent of damage caused by the 75 Gy dose was slightly greater than that by the 65 Gy dose (Figure 3a).

Histopathological analysis revealed a reduction in healthy air spaces, inflammatory cell infiltration, and collagen deposition in the 2-week and 6-week irradiated samples compared to that in non-irradiated tissues, regardless of the radiation dose applied (Figure 3b). Quantitatively, these criteria, especially the latter two, were higher in the 2-week irradiated tissues than in the 6-week irradiated tissues (Figure 3c–e). Overall, these findings indicate the successful establishment of the rodent model using either of the radiation doses. However, we observed some differences between the two irradiated tissue groups. For instance, foamy macrophages were only present in the slides from the 75 Gy-irradiated group, particularly in the 6-week group (Figure 3b). Additionally, higher numbers of infiltrated inflammatory cells and increased collagen deposition were observed in the 75 Gy-irradiated tissues compared to those in their 65 Gy counterparts. This dose-dependent difference was particularly evident in the collagen deposition of the 2-week irradiated tissues (Figure 3e).

### 2.4. Comparative Transcriptomic Analysis between 65 Gy and 75 Gy Models

After sequencing and filtering the total RNA of 56 irradiated lung samples, the obtained clean reads accounted for more than 98.5% of the raw reads, indicating the high quality of the acquired reads and suitability for further analysis. Notably, replicate no. 2 of the 6-week 65 Gy-irradiated tissues showed reduced reads in both raw and processed forms. Furthermore, it exhibited the lowest mapping ratio. However, the rest of the RNA-seq and alignment data generally exhibited high enough quality to conduct the downstream analyses. Detailed RNA-seq and alignment data from 65 Gy-irradiated lungs can be found in Table 1.

Examining the Pearson correlation, two major clusters were observed, distinguishing between non-irradiated and irradiated groups, again indicating radiation-induced transcriptional differences regardless of the IR dose. Furthermore, each cluster was further sub-clustered to accurately distinguish between the 2-week and 6-week samples, regardless of the radiation dose (Figure 4a). Overall, these results again confirm the molecular distinction between the inflammation and fibrosis stages. Furthermore, they suggest a partial similarity in the transcriptional response between 75 Gy and 65 Gy during RILI development.

Following the calculation of DEGs among the transcripts of the 65 Gy-irradiated groups, we observed higher DEGs in the fibrosis stage than in the inflammation stage. Moreover, there were more upregulated genes than downregulated genes at both stages (Appendix A), which was also observed in 75 Gy-irradiated groups. Comparing the 65 Gy-irradiated mice with 75 Gy-irradiated counterparts, the number of DEGs in the 65 Gy-irradiated groups was consistently lower than that in the 75 Gy-irradiated groups, regardless of the time point, indicating a greater impact of 75 Gy IR dose at the transcriptional level (Appendix A).

Further analysis of biological processes and pathway enrichment demonstrated time-dependent clustering. Overlapping and distinct enrichments were observed between the 2-week and 6-week clusters. For example, inflammatory and immunological responses, as well as immune cell activation, were predominantly represented in all groups, while the mitotic cell cycle process and immunoglobulin production were specifically enriched in the 2-week and 6-week groups, respectively. Additionally, certain enrichments were higher in the inflammation group than in the fibrosis group, such as the response to interferon-beta. Regarding differences between the 65 Gy- and 75 Gy-irradiated groups, both exhibited similar enriched terms regardless of the stage.

However, the 75 Gy-irradiated group showed more significant enrichment at 2 weeks than the 65 Gy-irradiated group. Moreover, the terms myeloid leukocyte activation and response to interferon-gamma were more enriched in the 75 Gy-irradiated group at 2 weeks compared to those in the other groups (Figure 4b).

### 2.5. Validation of Similarities and Differences between 65 Gy and 75 Gy Models by Reverse Transcription Quantitative PCR (RT-qPCR)

To validate the findings from our GO and KEGG analyses, we performed RT-qPCR to confirm the expression of selected DEGs. The top DEGs with the highest fold changes and significant *p*-values were chosen for this analysis, representing different categories of DEGs. The selection was based on Venn diagram analysis, which identified common and dose-specific DEGs in the four groups (Appendix A). We selected specific genes, namely *SAA3* and *CXCL10*, to represent the intersected DEGs that generally responded to radiation (IRintDEGs). *ACOD1* and *SKA1* were chosen as representatives of the 2-week-specific DEGs related to inflammation (inflammation-intersected DEGs, IintDEGs). Furthermore, *ADIPOQ*, *CAR3*, and *PLIN1* were selected as representatives of the 2-week 75 Gy-specific DEGs (2w75sDEGs). Unfortunately, due to low RNA yield in the 6-week groups, which limited our ability to perform RT-qPCR analysis, we were unable to examine the expression of the top DEGs representing the 6-week groups. A list of genes and their corresponding RT-qPCR primers can be found in Appendix A.

As shown in Figure 5a, the expression of *SAA3*, *CXCL10*, *ACOD1*, and *SKA1* was upregulated in the irradiated groups compared to that in the non-irradiated groups. Additionally, the expression of *ADIPOQ*, *CAR3*, and *PLIN1* was significantly suppressed in the 75 Gy-irradiated group, while no significant changes were observed in the 65 Gy-irradiated group.

To assess the correlation between the fold change in DEGs obtained from RNA-seq and RT-qPCR results, we performed a correlation analysis. The results indicated a high correlation between the two datasets, with *R*² values of 0.96 and 0.98 for the 75 Gy- and 65 Gy-irradiated groups, respectively (Figure 5b). This high correlation suggests the reliability of our downstream analysis and strengthens the confidence in our findings.

## 3. Discussion

We previously established a focally applied high-radiation-dose animal model based on a radiation dose of 75 Gy. In this study, we aimed to explore the associated transcriptomic signature. In addition, we aimed to optimize the high-dose ionizing radiation rodent model by lowering the radiation dose. Previous studies analyzing the transcriptome of classic RILI models have consistently identified inflammatory response, macrophage activation, cell cycle, DNA damage, mitotic division, and extracellular matrix (ECM) remodeling as the most representative biological processes and pathways [5,10,11].

Transcriptomic data analysis after a radiation dose of 75 Gy revealed DEGs encoding several proteins known to serve as biomarkers for inflammation and fibrosis stages. These proteins included TGF-β-induced protein, TNF and its associated proteins, granulocyte–macrophage colony-stimulating factors and their associated proteins, IL-4 and its associated proteins, IL-1a, PDGF, and IL-6. This initial finding presents a promising opportunity for utilizing the data in subsequent investigations. Downstream analysis of the transcriptomic data indicated a similar GO enrichment pattern, which was validated via RT-qPCR analysis. During the early stages of RILI, the exposure of normal tissue to radiation induces the production of reactive oxygen species (ROS), leading to DNA damage, impaired cell division, and eventually, cell apoptosis. In the later stages, when macrophage activation is dysregulated due to suppressed anti-inflammatory processes, pro-inflammatory cytokines are overproduced, which triggers fibroblast proliferation, activation, and differentiation into myofibroblasts, contributing to tissue fibrosis [12]. Therefore, the enrichment of biological processes and pathways associated with DNA damage and the cell cycle during the early stages and with macrophage activation, immunoglobulin production, and ECM remodeling during the later stages reflects these cellular processes. Notably, immunoglobulin production has not been observed in previous transcriptomic explorations of classic RILI models. However, experimental observations have indicated that secretory immunoglobulin induces fibroblast proliferation, activation, differentiation into myofibroblasts, and collagen production, similar to the effects of TGF-β1 [13]. This emphasizes the role of immunoglobulins in the induction of the fibrosis stage of the high-dose RILI model.

We conducted a comparative diagnostic study to evaluate the suitability of 65 Gy and 75 Gy as radiation doses for future preclinical studies. Our investigation involved CT imaging, histopathological analysis, and molecular assessments. We focused on the clinical impact of 75 Gy, particularly its effect on lung volume reduction, using micro-CT imaging. The results showed a reduction in lung volume of approximately 15–20% at 6 weeks post-irradiation, consistent with previous findings. Comparatively, the lung volume reduction observed at 65 Gy was lower than that at 75 Gy. This discrepancy suggested that the higher dose of 75 Gy might lead to excessive damage. Histopathological analysis supported the micro-CT observations, revealing more fibrotic features in the tissues exposed to 75 Gy, especially during the inflammation stage. Notably, in the 2-week 65 Gy-irradiated tissues, the lesions in the targeted area were not clearly visible, unlike in the 75 Gy-irradiated tissues in our study and previous studies [14]. Additionally, the presence of foamy macrophages, a characteristic feature associated with high-dose radiation, was only observed in the 6-week 75 Gy-irradiated tissues [15]. The presence of foamy macrophages is known to exacerbate pulmonary inflammation and fibrotic reactions by disrupting the pulmonary lipid clearance process during fibrosis progression [16].

Considering these observations, we performed protein–protein interaction (PPI) analysis specifically on the DEGs specific to the 6-week 75 Gy group (referred to as 6w75sDEGs), which were identified using a Venn diagram (Appendix A). The analysis revealed an enrichment of lipid metabolism-related processes, including arachidonic acid metabolism and eicosanoid metabolic processes, both of which have been linked to lung fibrosis progression [17,18,19]. Therefore, we concluded that 75 Gy, unlike 65 Gy, impaired lipid metabolic processes, leading to an exacerbated fibrotic reaction.

Comparative transcriptomic analysis also revealed an overlapping enrichment pattern similar to that mentioned above, except for ECM remodeling. However, when we specifically examined the DEGs using a Venn diagram and focused on the intersection of inflammation- and fibrosis-related DEGs (referred to as IintDEGs and FintDEGs, respectively), we found ECM-related terms enriched in the latter (Appendix A). Nonetheless, we also observed increased enrichment across all biological process and pathway terms in the 75 Gy-irradiated group during the inflammation stage, which corresponded to the excessive fibrosis observed in the histological analysis of the 75 Gy-irradiated 6-week samples. Interferon-gamma (IFN-γ) plays an unfavorable role in RILI progression by inhibiting Th2-driven pro-inflammatory cytokines such as IL-4 and IL-13 through its anti-inflammatory effects [20]. The term ”response to interferon-gamma” was predominantly enriched in the 75 Gy 2-week group compared with that in the other groups. Additionally, the Venn diagram confirmed that this term was exclusively enriched in the 75 Gy 2-week group, specifically in the 2w75sDEGs (Figure 5), with most of the top DEGs, including *ADIPOQ* and *PLIN1*, being downregulated (Appendix A). The *ADIPOQ* gene encodes adiponectin, an adipokine secreted by adipocytes that plays a crucial role in regulating various cellular processes associated with obesity, including lipid metabolism and insulin sensitivity [21]. While *ADIPOQ* is generally upregulated in cystic lung fibrosis, further investigations have revealed that its upregulation is only observed in cases with mild symptoms, suggesting a suppressive effect on fibrosis progression [22]. This effect has been demonstrated in studies utilizing adiponectin as a therapeutic agent for different types of fibrosis, such as eye fibrosis and paraquat-induced lung fibrosis [23,24]. Although the function of adiponectin contradicts the action of INF-γ in diabetes [25], the relationship between adiponectin and INF-γ in the context of lung fibrosis is still unclear.

The *PLIN1* gene encodes perilipin 1, a protein primarily involved in lipid metabolism by regulating lipolysis in adipocytes [26,27]. Lipids have been shown to exacerbate the progression of lung fibrosis [28], and inhibiting lipolysis leads to an increase in dermal ECM deposition [29]. Additionally, studies have reported that *PLIN1*-knockout mice exhibit sustained inflammation due to increased storage of lipid droplets in adipocytes [30]. Increased expression of INF-γ is associated with elevated mRNA levels of genes involved in lipid droplet formation and storage, including perilipins [31]. Consequently, it is expected that the impairment of INF-γ-mediated lipid metabolism, which plays a role in controlling fibrosis, by the 75 Gy radiation dose would result in an exacerbated fibrotic reaction.

Considering the significance of utilizing our preclinical rodent RILI model for evaluating potential therapeutic interventions during the inflammation stage, these findings suggest that the 65 Gy- and 75 Gy-based model can be appropriate for assessing anti-RILI candidates.

## 4. Materials and Methods

### 4.1. Animal Experiments

Animal grouping and radiation experiments were conducted following previously described protocols [32] with minor modifications. The protocols involving animals were reviewed and approved by the Animal Care and Use Committees of Yonsei University Medical School (YUHS-IACUC; 2021-0041) and strictly adhered to the relevant guidelines. Wild-type male C57BL/6 mice (*Mus musculus*) were obtained from Charles River Korea (Orient Bio, Seongnam, Republic of Korea) at the age of 6 weeks. Prior to the initiation of the irradiation experiment, mice were housed in cages (five mice per cage) for a week to allow for acclimatization. The irradiation system was delivered using X-RAD 320 (Precision, North Branford, CT, USA) with a tungsten anode. The developed image-guided focal irradiation system consists of a charge-coupled camera (650D, Canon Inc., Tokyo, Japan). For accurate positioning, the fiducial markers that indicated the field center and the target were matched and investigated. The percentage depth doses (PDDs) were measured using a cylindrical ionization chamber within a solid water phantom (PTS, RW3) with a source-to-surface distance of 170 mm at 300 kV and 12.5 mA. As a result of the PDD measurement of 3.0 mm cone size, the dose rate was measured as 25.7 cGy/s at a depth of 8.0 mm. The left lung of each mouse was then exposed to a single fraction of radiation at a dose of 45, 55, 65, or 75 Gy using an image-guided small-animal irradiator (X-RAD 320, Precision, North Branford, CT, USA). The experimental mice were divided into three groups, with non-irradiated mice serving as the control group. Following irradiation, the mice were housed in cages and euthanized after 2 weeks to examine the inflammation stage or after 6 weeks to investigate the fibrosis stage. Anesthesia was induced by intraperitoneal administration of a mixture containing 80 mg/kg ketamine and 5 mg/kg xylazine.

### 4.2. Micro-CT Analysis

Micro-CT imaging was conducted as described previously [32] with some modifications. Images were obtained using a quantum GX2 micro-CT scanner (Perkin Elmer Inc. Waltham, MA, USA). The microfocus X-ray source used in this scanner was a tungsten anode. A fixed filter of 0.5 mm aluminum and 0.06 mm copper was placed in front of the exit port to remove low-energy X-rays that contributed to the dose but did not improve the image quality. Images were acquired using a respiratory-gated technique with the following parameters: X-ray tube voltage, 90 KV; X-ray tube current, 88 µA; and total scan time, 4 min. Volumetric analysis of CT images was also performed with Mayo Clinic Analyze 12.0 software (Biomedical Imaging Resource, Mayo Clinic, Rochester, MN, USA) using equally spaced coronal images.

### 4.3. Histopathological Analysis

The assessment of slides was conducted by pathologists at DKKorea Co., Ltd. (Seoul, Republic of Korea) using a dual-rate semi-quantitative method, as previously described [32]. Histological evaluation was performed on lung tissue sections, which were stained with hematoxylin and eosin and Masson’s trichrome to examine inflammation and fibrosis, respectively. The level of inflammation was estimated using the inflammation score [33], while fibrosis was assessed using either the modified Ashcroft score [34,35] or image-based analysis utilizing Image J software 1.53a [36]. A detailed description of the scoring criteria for each parameter can be found in Appendix A.

### 4.4. Extraction, Purification, and Sequencing of RNA

Total RNA was extracted and isolated from the lesions of irradiated tissues using QIAzol^®^ lysis reagent (QIAGEN, Hilden, Germany) according to the manufacturer’s instructions. The RNA library construction required for RNA-seq was independently prepared with 1 µg of finally prepared total RNA using the Illumina TruSeq Stranded mRNA sample prep kit (Illumina, Inc., San Diego, CA, USA). The process of library preparation began with mRNA enrichment, which involved isolating mRNA from total RNA using poly T-attached magnetic beads. Following this step, the collected mRNA was fragmented into small pieces, and cDNA was prepared using SuperScript II reverse transcriptase (Invitrogen, Thermo Fisher Scientific, Waltham, MA, USA). The cDNA fragments were ligated with adapters and enriched by PCR to obtain a cDNA library. The libraries were quantified using KAPA library quantification kits for Illumina sequencing platforms according to the qPCR quantification protocol guide (KAPA BIOSYSTEMS, #KK4854; Wilmington, MA, USA) and qualified using TapeStation D1000 ScreenTape (Agilent Technologies, #5067-5582; Santa Clara, CA, USA). Indexed libraries were then submitted to Illumina NovaSeq, and paired-end (2 × 100 bp) sequencing was performed by Macrogen Inc. (Macrogen, Seoul, Republic of Korea). The raw sequencing data of our RNA experiment are publicly available in the genome Gene Expression Omnibus database and can be accessed via the accession number GSE205025.

### 4.5. Whole-Genome Transcriptomic Analysis

The quality assessment of the obtained reads was conducted using FastQC v0.11.7. Reads with a quality score of 30 or higher were considered of good quality. Samples with analyzable reads were identified, ensuring that over 95% of the total reads met the chosen quality score threshold. To maintain high-quality reads, Trimmomatic 0.38 was employed to remove short or low-quality reads, as well as adapter contamination. Only long, high-quality reads were retained, with a threshold read length of 36 base pairs. The efficacy of the trimming process was validated by comparing the raw read QC before and after trimming using FastQC v0.11.7. For read mapping, the genome reference of *Mus musculus* (GRCm38.p6) and the annotation library (NCBI_108) were selected. The selected fragments were mapped to the reference genome using HISAT2 v2.1.0, which utilizes the Bowtie2 v3.4.1 aligner to handle spliced reads. The presence of a low percentage of unmapped reads (<1.5%) indicated the quality of the RNA samples and ensured the accuracy of subsequent analyses. Upon successful mapping, the read fragments were assembled to generate integrated transcripts, and gene expression levels were estimated using StringTie 2.1.3b.

### 4.6. Identification of DEGs

The detection of DEGs of the 65 Gy and 75 Gy irradiated groups, at 2 and 6 weeks, was conducted using the R package “edgeR 3.36.0” [37], in which filtered genes were normalized and statistical calculations of significant expression were performed using the trimmed mean of M-value and negative binomial methods, respectively. The resulting p-value was adjusted using the Benjamini–Hochberg method [38] to avoid falsely expressed genes. Up- and downregulated genes were expressed as positive or negative logarithm base 2 of the fold change (Log_2_FC), and |Log_2_FC| ≥ 1 and *p* ≤ 0.05 was set to be the threshold for significant DEGs. 

### 4.7. Expression and DEG Data Interpretation and Analysis

Following the acquisition of expression data and identification of DEGs, several downstream analyses were performed to investigate the impact of reducing the radiation dose on the transcriptional behavior of lung cells. To assess the variation among samples, MDS and the Pearson correlation matrix were computed and visualized. This was achieved using the R packages “mixOmics” [39] and “gplots” [40]. MDS provides a visual representation of the overall similarity or dissimilarity between samples, while the Pearson correlation matrix quantifies the strength and direction of the relationships between samples. To further analyze the DEGs, a Venn diagram was constructed. This allowed for the segregation of DEGs that were common to both the 75 Gy and 65 Gy radiation doses (intersected DEGs), as well as DEGs specific to each dose. The R package “VennDiagram” [41] was utilized for this purpose. To visualize the expression variation of the intersected DEGs, z-scored heatmap plots were generated. These plots provide a graphical representation of the expression of the intersected DEGs across samples, with z-scores indicating the relative expression. The “gplots” package [40] was employed to generate these heatmap plots.

### 4.8. Functional Enrichment, Network Analysis, and PPI Module Construction

To explore the enriched pathways and networks associated with the selected DEGs, the web-based gene-list analysis tool Metascape (http://metascape.org; accessed on 1 April 2023) was utilized with default parameters [42]. This tool enables comprehensive functional analysis of gene lists. In this analysis, the selected gene lists were inputted into Metascape, which utilized various genome resources for enrichment analysis. These resources included the KEGG pathway, GO biological processes, Reactome gene sets, canonical pathways, and CORUM [43]. All input genes were considered the background for enrichment analysis. Terms with a minimum count of >3 and a significance threshold of *p* < 0.01 were selected for further analysis. These terms were then grouped into clusters based on their similarity for subsequent testing. To account for multiple tests, q-values were calculated using the Benjamini–Hochberg procedure. Hierarchical clustering was performed on the enriched terms using Kappa scores as the similarity metric. Sub-trees with a similarity score exceeding 0.3 were defined as clusters. The most statistically significant terms within each cluster were identified as representative terms for their respective clusters. These terms provide insights into the common and unique biological processes, pathways, and networks associated with the selected DEGs.

### 4.9. RT-qPCR

Prior to sending extracted RNA samples for RNA-seq, aliquots were taken from the samples to verify the RNA-seq results using RT-qPCR. cDNA was immediately synthesized using a high-capacity RNA-to-cDNA^TM^ kit (Applied Biosystems, Thermo Fisher Scientific, Pleasanton, CA, USA) according to the manufacturer’s instructions. Briefly, 2 µg of total RNA was mixed with the synthesis mixture in a total volume of 20 µL. The synthesized cDNA was kept at −20 °C for subsequent real-time RT-qPCR. Primers for selected genes were determined using the webtool Primer3 using default settings, and the specificity was validated using the web-based primer-BLAST primer designing tool (https://www.ncbi.nlm.nih.gov/tools/primer-blast/; accessed on 20 June 2023). A list of primer pairs for each selected gene is provided in Appendix A. RT-qPCR was conducted in a 10 µL reaction mix consisting of 1 µL cDNA, 10 pmol of each primer pair, 5 µL 2X iQTM SYBR^®^ green supermix (Bio-Rad, Hercules, CA, USA), and water. The reaction was conducted using a CFX Connect^TM^ Real-Time System (Bio-Rad) with Optical Flat 8-Cap Strips (Bio-Rad).

## 5. Conclusions

As our research progresses towards the development of drugs for RILI, it becomes crucial to employ a more sophisticated disease model. We explored various endpoints, including molecular signatures, to determine if the previously employed high radiation doses in disease models, which focused on phenomena and mechanisms, could be slightly reduced while ensuring efficient experimental research within a limited timeframe.

Consequently, we conducted experiments with reduced radiation doses and identified 65 Gy as the optimal dose that induces inflammation and fibrosis in a well-scheduled manner, as confirmed through a series of imaging, histopathological, and molecular analyses. Additionally, through comprehensive comparative observations between models based on 65 Gy and 75 Gy, we discovered that the application of 75 Gy resulted in accelerated fibrosis, likely due to disruption of lipid metabolism processes.

The findings of this study serve as a significant milestone towards the establishment of robust preclinical therapeutic strategies for RILI using the dose 65 Gy, particularly for targeting early pathogenic events. Future work will concentrate on evaluating several preclinical candidate molecules for treating RILI utilizing the outcomes of this study. 

## Figures and Tables

**Figure 1 ijms-25-03766-f001:**
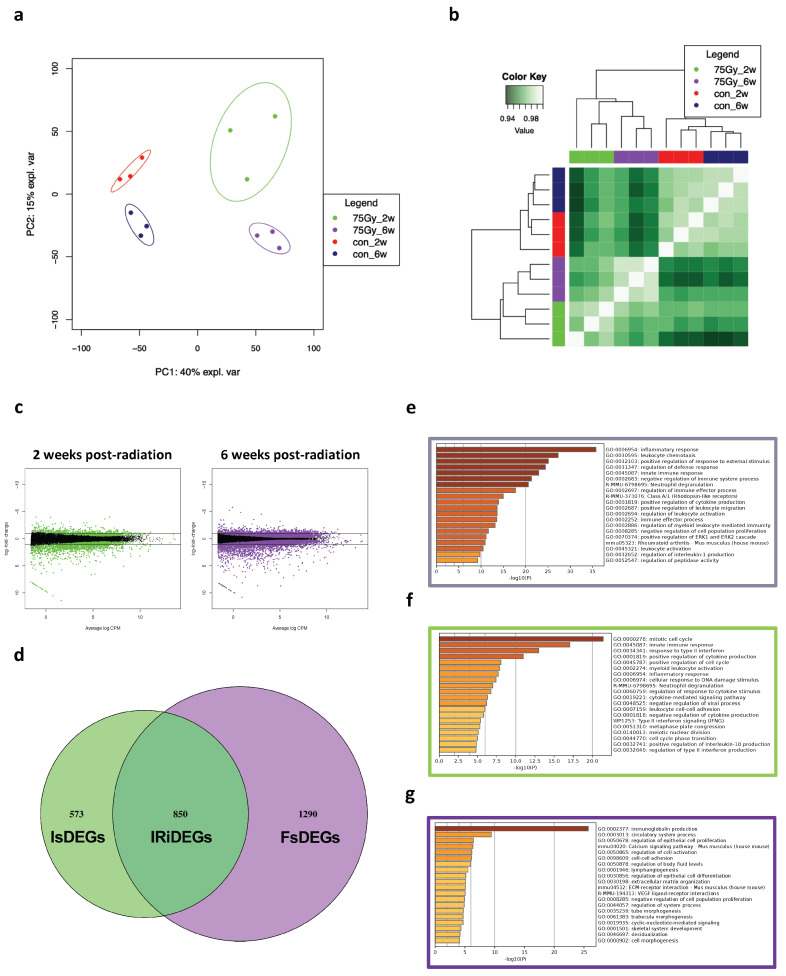
Genome−wide transcriptional differences between inflammation and fibrosis stages of 75 Gy-irradiated lung tissues. (**a**) Multidimensional scaling (MDS) and (**b**) heatmap correlation plot. Weakly expressed genes were removed, and a total of 21,355 genes were normalized and used for visualized correlations. The MDS and Pearson correlation showed a time-oriented correlation between the samples, in which 2-week and 6-week samples are separately clustered. (**c**) Mean difference (MD) plots showing log-fold change and average abundance of each gene at the inflammation and fibrosis stages. The black line is a threshold to distinguish genes with a log-fold change greater than 1. (**d**) Venn diagram showing intersected and exclusive DEGs between the two stages. (**e**–**g**) Bar plots representing GO enrichment analysis of inflammation-specific, IR-induced, and fibrosis-specific DEGs, respectively. Abbreviations: IsDEGs, inflammation-specific DEGs; IRiDEGs, ionizing radiation-induced DEGs; FsDEGs, fibrosis-specific DEGs.

**Figure 2 ijms-25-03766-f002:**
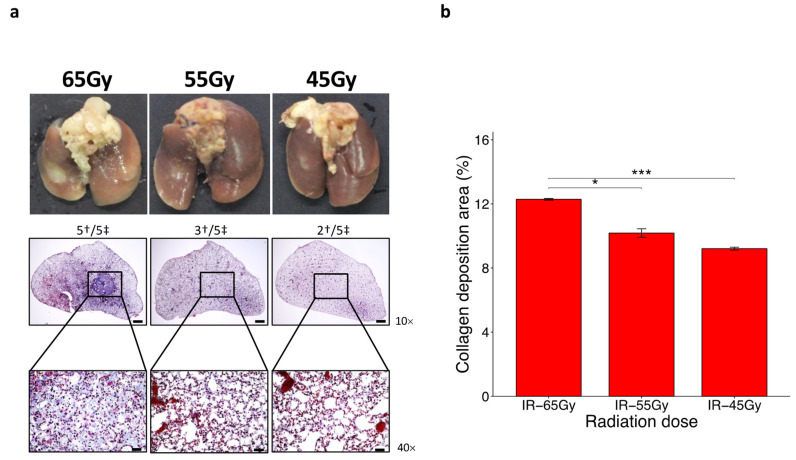
A dose of 65 Gy is the lowest radiation dose able to induce inflammation and fibrosis within 2 months. (**a**) Gross findings (upper panel) and Masson’s Trichome staining (lower panel) of lung sections after 6 weeks of focal irradiation at doses of 45, 55, and 65 Gy. Magnifications, 10× (scale bar is 1 mm), 40× (scale bar is 100 µm). (**b**) Percentage of collagen deposition of 6-week 45, 55, and 65 Gy-irradiated lung tissues. † and ‡ refer to damaged and totally irradiated lungs, respectively (n = 5, *t*-test, * *p* < 0.05, *** *p* < 0.001).

**Figure 3 ijms-25-03766-f003:**
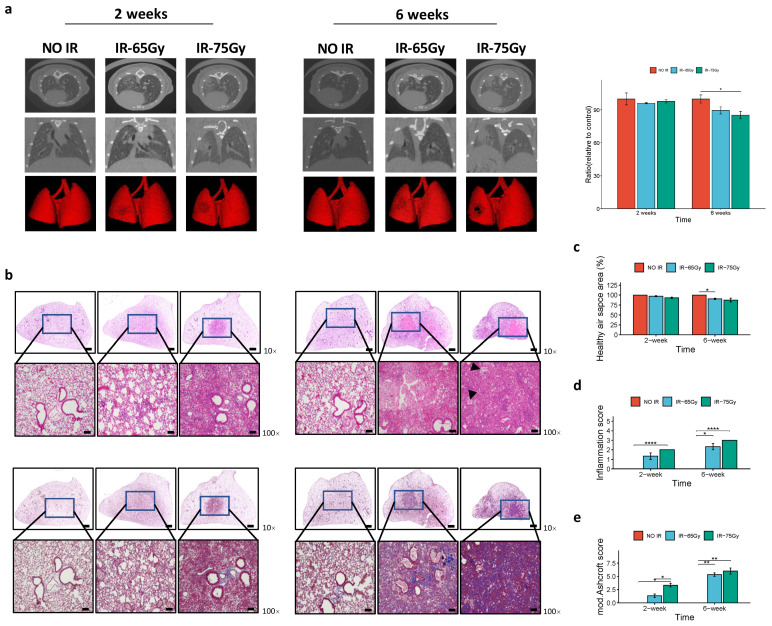
A radiation dose of 65 Gy induces lung injury with moderate damage at the inflammation stage. (**a**) Micro-computed tomography (CT) imaging (left panel). Representative micro-CT images of lung from control, 65 Gy-, and 75 Gy-irradiated mice, 3D horizontal and trans-axial images. Quantitative measurement of total normal lung volume of irradiated and control mice, showing a slightly significant greater amount of damage caused by 75 Gy (right panel). Lower volume indicates more lung damage by ionizing radiation. (**b**) Hematoxylin and eosin staining (upper panel) and Masson’s Trichome staining (lower panel) of lung sections after 2 weeks (left panel) and 6 weeks (right panel) of focal irradiation at 65 Gy and 75 Gy. Arrow refers to the foamy macrophages. Magnification, 10× (scale bar is 1 mm), 100× (scale bar is 100 µm). Percentages of healthy air space area (**c**), inflammation scores (**d**), and fibrosis scores (**e**) of 2-week and 6-week 65 Gy- and 75 Gy-irradiated lung tissues (n = 3, *t*-test, * *p* < 0.05, ** *p* < 0.01, **** *p* < 0.0001).

**Figure 4 ijms-25-03766-f004:**
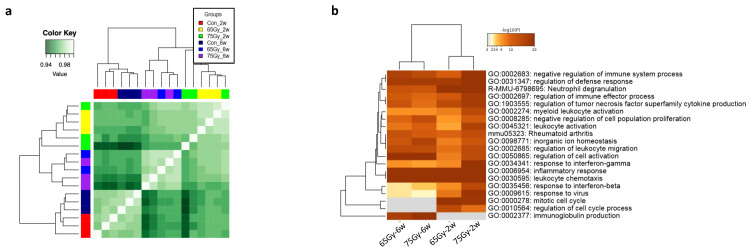
Genome-wide transcriptional variations between 65 Gy and 75 Gy at inflammation and fibrosis stages. (**a**) Heatmap correlation plot. Poorly expressed genes were removed, and a total of 21,355 genes were normalized and used for visualized correlations. Pearson correlation showed a time- and IR dose-oriented correlation between the samples, in which 2-week and 6-week samples are separately clustered. (**b**) Enrichment heatmap of clustered GO terms and KEGG pathways showing clustering of 2-week groups separately from 6-week counterparts, irrespective of the radiation dose.

**Figure 5 ijms-25-03766-f005:**
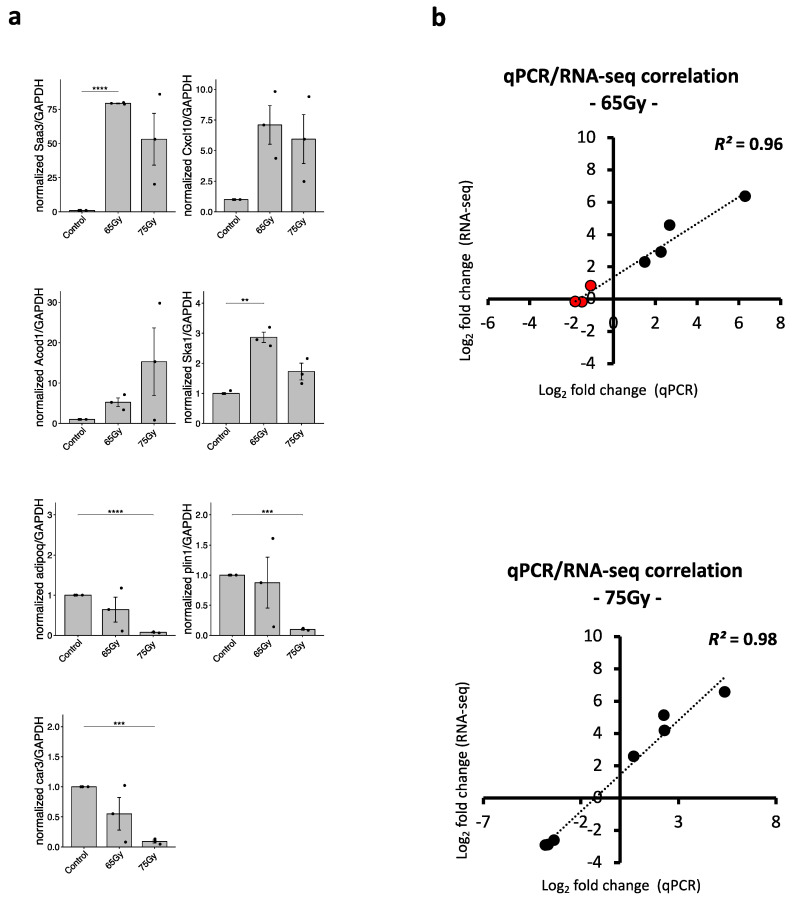
Validation of RNA-seq analysis using RT-qPCR. (**a**) Relative expression analysis of top expressed genes. (**b**) Linear correlation between log2 FC of RNA-seq and RT-qPCR of top expressed genes in 65 Gy- (upper panel) and 75 Gy- (lower panel) irradiated 2-week samples. Red points indicate the genes *ADIPOQ*, *PLIN1*, and *CAR3*, which are not DEGs identified in the 65 Gy-irradiated tissue, but whose expression was investigated by RT-qPCR. Abbreviations: FC, fold change (Wilcoxon test, ** *p* < 0.01, *** *p* < 0.001, **** *p* < 0.0001).

**Table 1 ijms-25-03766-t001:** Statistical analysis of RNA-seq quality assessment and alignment coverage of non-irradiated, 65 Gy-, and 75 Gy-irradiated lung tissues.

Time	Group	Replicate	RNA-Seq Data				Alignment Data	
			Raw Reads	Trimmed Reads	Raw Reads vs. Trimmed Reads	GC Content (%)	Total Mapped	Total Unmapped
2 weeks	Control	1	66,847,010	66,073,590	98.84	48.79	65,182,623 (98.65%)	890,967 (1.35%)
		2	63,021,390	62,284,648	98.83	49.34	61,617,863 (98.93%)	666,785 (1.07%)
		3	72,993,584	72,081,246	98.75	49.53	71,293,386 (98.91%)	787,860 (1.09%)
	65 Gy	1	79,873,572	78,875,404	98.75	49.60	78,075,192 (98.99%)	800,212 (1.01%)
		2	82,604,278	81,604,686	98.79	49.40	80,581,993 (98.75%)	1,022,693 (1.25%)
		3	73,278,816	72,414,130	98.82	49.56	71,653,924 (98.95%)	760,206 (1.05%)
	75 Gy	1	77,172,286	76,155,556	98.68	49.17	75,122,624 (98.64%)	1,032,932 (1.36%)
		2	89,558,428	88,540,584	98.86	49.21	87,557,242 (98.89%)	983,342 (1.11%)
		3	76,282,682	75,317,680	98.73	49.49	74,579,006 (99.02%)	738,674 (0.98%)
6 weeks	Control	1	85,460,960	84,448,858	98.82	49.27	83,548,257 (98.93%)	900,601 (1.07%)
		2	77,140,466	76,145,872	98.71	49.44	75,236,435 (98.81%)	909,437 (1.19%)
		3	78,125,372	76,988,642	98.54	49.36	76,054,270 (98.79%)	934,372 (1.21%)
	65 Gy	1	80,695,798	79,595,842	98.64	49.56	78,625,912 (98.78%)	969,930 (1.22%)
		2	60,530,124	59,600,646	98.46	50.35	58,558,124 (98.25%)	1,042,522 (1.75%)
		3	77,818,676	76,762,422	98.64	49.33	75,816,780 (98.77%)	945,642 (1.23%)
	75 Gy	1	82,035,406	80,945,774	98.67	49.26	79,905,992 (98.72%)	1,039,782 (1.28%)
		2	72,804,658	72,064,966	98.98	48.55	71,335,395 (98.99%)	729,571 (1.01%)
		3	73,353,186	72,494,140	98.83	48.75	71,765,561 (98.99%)	728,579 (1.01%)

## Data Availability

The raw sequencing data of our RNA experiment are publicly available in the genome database Gene Expression Omnibus (GEO) under the accession number GSE205025. Other data supporting the present study are available from the corresponding author upon reasonable request.

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
