# Peer review of "Transcriptional Changes in Radiation-Induced Lung Injury: A Comparative Analysis of Two Radiation Doses for Preclinical Research"

_ijms, 2024, doi:10.3390/ijms25073766_

Round 1

Reviewer 1 Report

Comments and Suggestions for Authors

In this manuscript, Farh and colleagues report on a study, which tested the effect of focal irradiation of murine lung tissue with different dosis, ranging from 45Gy to 75Gy, at the level of histology and transcriptome. Based on previous results and their current findings the authors suggest that 65Gy represents the optimal dose for studying radiation induced lung injury in mice.

The paper is well written and easy to understand. Nevertheless several aspects need clarifications:

+ page 3 and Figure 1b: it is claimed that irradiated samples cluster in a time dependent manner. Yet, looking at the dendrogram of the heatmap depicted in Fig. 1b. only two of the samples 75Gy_2w belong to the same cluster at the first level.  A similar situation is encountered in Figure 4a, where only two of the Con_2w group in one cluster; in both figures the leftmost sample is not part of the cluster at the first level. In addition the legend to Figure 1 talks about  4-week instead of six weeks.

+ page 3: it is stated that Cytoscape was used for overrepresentaton analysis, while the Material&Methods section mentions Metascape

+ Figure 4b: Several GO categories can include the very same genes and thus can not be seen as independent from each other.  In this way, some differentially expressed genes have more weight than others when doing the clustering. What strategy was employed to avoid the introduction of clustering bias due to the overlap and partly redundance of GO categories.

+ it would be interesting to learn about the authors opinion to what extent the observed changes of gene expression patterns are attributed to irradiation induced changes of cell type composition versus irradiation induced gene expression changes in a given cell type

+ page14: FASTQC was used to determine the quality of sequence reads. Given the rather high numbers of sequence reads per sample (which is usually a good thing), it would be interesting to learn about the sequence duplication levels to get an idea of library complexity.

+ page 14: it is stated that filtering used a threshold p=<0.5. Is that p-adjust?

+ data availability: the gene expression data produced in this study can be useful for all researchers interested in irradition induced fibrosis. The authors state that RNASeq data have been uploaded to GEO, but no reviewer token is provided. Thus I can confirm that the accession number exists on GEO, but I could not check what data can be accessed via this number. Also, please ensure that data are released as soon as the manuscript is accepted (currently scheduled to be released: 27. May 2025)

Reviewer 2 Report

Comments and Suggestions for Authors

The authors developed the radiation-induced lung injury (RILI) model in C57BL mice with high doses of irradiation, in which the fibrosis occurs within 2 months. In the present study, the global gene expressions were analyzed in the RILI by the RNA-seq, which identified the DEGs both in 2 and 6 weeks after the exposure. The effects of 75 Gy were compared to those of 65 Gy. Data from the genome-wide expression analysis in the RILI are interesting. However, the purposes and the conclusions of the present study are not clearly described.

1. Understanding the value of the DEGs data between the control and 75 Gy, the discussion is somewhat vague. Did the present data confirm the previously known gene expression changes after the radiation? For instance, what did the present analysis find in the gene expression changes in the Smad/TGFb pathway and the related genes that are important in developing RILI? What about IL-6 and other genes in the authors’ previous paper?

2. In Introduction, the purpose of the present study is not clear. Is it the optimization of the radiation dose of the author’s mouse model? Or, Is it the identification of critical genes/pathways in RILI?

3. Why did the authors need to examine 65 Gy or lower instead of 75 Gy? Were there any problems with the original 75Gy model, so that the authors needed to find the ‘optimum’ dose?

4. The title ‘Radiation dose-dependent....’ does not sound scientific, since the author compared only two different doses. The title needs to be changed to reflect the main purpose of the present study.

Round 2

Reviewer 1 Report

Comments and Suggestions for Authors

Many thanks for carifying my open questions.